# Analysis of the Potassium-Solubilizing *Priestia megaterium* Strain NK851 and Its Potassium Feldspar-Binding Proteins

**DOI:** 10.3390/ijms241814226

**Published:** 2023-09-18

**Authors:** Xinyue Wu, Zijian Zhao, Zirun Zhao, Youjun Zhang, Mingchun Li, Qilin Yu

**Affiliations:** 1Key Laboratory of Molecular Microbiology and Technology, Ministry of Education, Department of Microbiology, College of Life Sciences, Nankai University, Tianjin 300071, China; 13956130205@163.com (X.W.); 2010303@mail.nankai.edu.cn (Z.Z.); 2120221390@mail.nankai.edu.cn (Z.Z.); nklimingchun@163.com (M.L.); 2School of Environmental Science and Engineering, Tianjin University, Tianjin 300072, China; zhangyoujun001@126.com

**Keywords:** *Priestia megaterium*, potassium-solubilizing bacterium, genome sequencing, potassium feldspar

## Abstract

Potassium-solubilizing bacteria are an important microbial group that play a critical role in releasing mineral potassium from potassium-containing minerals, e.g., potassium feldspar. Their application may reduce eutrophication caused by overused potassium fertilizers and facilitate plants to utilize environmental potassium. In this study, a high-efficiency potassium-solubilizing bacterium, named NK851, was isolated from the *Astragalus sinicus* rhizosphere soil. This bacterium can grow in the medium with potassium feldspar as the sole potassium source, releasing 157 mg/L and 222 mg/L potassium after 3 days and 5 days of incubation, respectively. 16S rDNA sequencing and cluster analysis showed that this strain belongs to *Priestia megaterium*. Genome sequencing further revealed that this strain has a genome length of 5,305,142 bp, encoding 5473 genes. Among them, abundant genes are related to potassium decomposition and utilization, e.g., the genes involved in adherence to mineral potassium, potassium release, and intracellular trafficking. Moreover, the strong potassium-releasing capacity of NK851 is not attributed to the acidic pH but is attributed to the extracellular potassium feldspar-binding proteins, such as the elongation factor TU and the enolase that contains potassium feldspar-binding cavities. This study provides new information for exploration of the bacterium-mediated potassium solubilization mechanisms.

## 1. Introduction

Potassium is one of the most important plant nutrients and is essential for plant growth, metabolism, and development [1,2]. Potassium is the most abundant cation in living plant cells and plays an important role in osmotic adjustment, protein synthesis, membrane polarization control, carbohydrate metabolism, and enzyme activation [3,4,5,6]. From the perspective of plant nutrition, potassium can be divided into mineral potassium, slowly available potassium, and available potassium. Among them, only available potassium can be absorbed and utilized by plants, but its relative content in soil is only 1~2% [7,8]. Therefore, artificial intervention is needed in agricultural production to meet the potassium demand of crops. At present, chemical potash fertilizers are widely used in agricultural production to improve the contents of available potassium in soil. However, indiscriminate abuse of potash fertilizers reduces soil pH, increases soil exchangeable base, causes a lot of soil pollution and degradation, and further reduces productivity [9,10,11]. The latest data from the Food and Agriculture Organization of the United Nations (FAO) shows that 20–25% of the land in the world is degraded every year [12]. Therefore, people are committed to releasing the existing mineral potassium in the soil in a sustainable and green way [13].

Potassium-solubilizing bacteria are a class of bacteria that decompose potassium-containing inorganic minerals, e.g., potassium feldspar. They can dissolve potassium-containing minerals through multiple processes: (a) Acidolysis: The organic acids (tartaric acid, citric acid, etc.) released by potassium-solubilizing bacteria reduce soil pH, protonate potassium-containing minerals, and thus release potassium [8,14]. (b) Indirect solubilization: The organic acids secreted by potassium-solubilizing bacteria can chelate Si^4+^, Mg^2+^, and Ca^2+^ complexed with K^+^ in the minerals and indirectly dissolve potassium [15]. (c) Synthesis of biofilm: On the surface of the potassium-containing minerals, bacterial cells secret various extracellular macromolecules (e.g., proteins and polysaccharides) to form biofilms, in which the interaction between bacteria and macromolecules contributes to weathering of minerals and dissolution of potassium [16]. However, the kinds of the macromolecules involved in potassium release remain to be investigated.

*Priestia megaterium* is a widely distributed bacterium in nature, e.g., in soil, plant seeds, potato tubers, etc. The existing research shows that this species may promote the growth of plants from many aspects, including optimizing the structure of plant rhizosphere microbial community, regulating the expression of plant hormones [17,18], reducing the accumulation of harmful substances in plants under salt stress and drought stress [19], improving the resistance of plants to heavy metals [20], and inducing systemic resistance to inhibit the invasion, development, and reproduction of pests such as *Heterodera glycines* [21,22].

In this study, we isolated the *P. megaterium* strain NK851 from the *Astragalus sinicus* rhizosphere soil. Potassium-releasing analysis exhibited the high efficiency of potassium solubilization from potassium feldspar. Genome sequencing and analysis and protein mass spectrometry were further performed to investigate the mechanism of potassium solubilization by this bacterium. This study provides a theoretical basis for further exploration of potassium-solubilizing bacteria as microbial fertilizers widely used in agriculture.

## 2. Results

### 2.1. Isolation and Characterization of the P. megaterium Strain NK851

To screen high-efficiency potassium-solubilizing bacteria, the *Astragalus sinicus* rhizosphere soil was selected for isolation of pure strains on the potassium-solubilizing medium agars. A series of bacterial strains were finally obtained, i.e., NK878, NK879, NK850, NK851, NK852, NK853, NK854, NK855, NK856, and NK657. After incubation in the strains and the liquid potassium-solubilizing medium for 3 days, all of the ten strains induced obvious potassium release from potassium feldspar to 37~156 mg/L (Figure 1a). Especially, the strain NK851 induced the highest levels of potassium release (156 mg/L, Figure 1a). As the culture time of NK851 increased, the released potassium levels slightly increased to 7 mg/L after 1 day and then remarkably increased to 127 mg/L after 2 days, reaching 222 mg/L after 5 days (Figure 1b). Morphological monitoring of NK851 by SEM revealed that the strain had a rod-like morphology with round and blunt ends, linking into chains in liquid LB medium (Figure 1c). Furthermore, during culturing of NK851 in the potassium-solubilizing medium, the bacterial cells closely adhered to or bound potassium feldspar (Figure 1d), indicating their strong interaction with the minerals.

### 2.2. Phylogenetic Analysis of NK851

In order to assure the taxonomic position of NK851, BLAST analysis was performed based on the sequencing results of NK851 16S rRNA (Accession number PRJNA1008360). The results indicated that its 16S rDNA sequence has the highest similarity with those of the *P. megaterium* ATCC 14581 (98.83%) and the *P. megaterium* IAM 13418 (98.64%). We further constructed the phylogenetic tree of NK851 using the MEGA11.0 software via the neighbor-joining method (Figure 2). The results showed that NK851 was closely related to *P. megaterium*. Therefore, it could be confirmed that the potassium-solubilizing strain NK851 belonged to *P. megaterium*.

### 2.3. Genome Assembly and Analysis of NK851

To further characterize the *P. megaterium* NK851, its genome DNA was fully sequenced, followed by genome assembly and functional analysis. The whole genome of the strain consists of a round chromosome and four round plasmids. The chromosome is 5,305,142 bp in length, with a GC content of 37.99%, and the total length of the coding genes is 4,434,840 bp, containing 5473 genes, accounting for 83.6% of the total chromosome length. The mean gene length is 810 bp, and the N90 scaffold size is 5,305,142 bp. The size of the repeat sequences is 11,583 bp, representing 22% of the whole genome. Non-RNA-encoding genetic prediction indicates that the genome contains 42 rRNA and 116 tRNA. There are 11 clustered regularly interspaced short palindromic repeated (CRISPR) sequences and 4 gene islands in the genome.

Based on the known whole-genome information, the circular genome map of NK851 was constructed (Figure 3). To dissolve and release potassium ions necessary for growth and reproduction from mineral potassium feldspar, NK851 needs to coordinate a series of genes for interaction. Under the stimulation of the external environment, multiple sets of intracellular signaling pathways are activated to regulate the expression of key genes related to potassium release and absorption, e.g., (a) the genes involved in adherence to mineral potassium, such as cell wall/membrane/envelope biosynthesis genes (Figure 3M), in which 173 genes are predicted; (b) the genes that function in potassium release, such as amino acid transport and metabolism genes (Figure 3E), in which 407 genes are predicted; (c) the genes related to intracellular trafficking, secretion, and vesicle transport (Figure 3U), which can transport the synthesized potassium-solubilizing substance to the outside of the cell; (d) the genes involved in inorganic ion transport and metabolism (Figure 3P), which can transport the released potassium ions into the cells for their utilization; and (e) the genes required for energy production and conversion, with 254 genes predicted in total (Figure 3C). Therefore, NK851 has abundant genes related to potassium decomposition and utilization, demonstrating its efficient potassium solubilization ability at the molecular level.

The NK851 protein sequences were compared with the *Non-Redundant Protein Database* (*NR* database) in BLAST, and the corresponding species information was obtained from the annotation database. A total of 5314 protein-encoding genes were annotated. Combined with the comparison results of phylogenetic trees, it was finally concluded that NK851 has the highest similarity with the *P. megaterium* reference strain 22-2 (Reference Genome Number ASM993541v1, 39.46%, Figure 4a), and it was determined as *P. megaterium*.

The *Gene Ontology* (*GO*) database was then applied to annotate genomic information. The genetic functions annotated by the *GO* database include cellular components, fractional functions, and biological processes. A total of 4010 genes were annotated in the database. In the cell components, the membrane-associated protein-coding genes were the most abundant (~34.3%). In the molecular function, the content of protein-coding genes related to catalysis (~57.5%) and binding (~36.9%) was high. In biological processes, the protein-coding genes related to metabolism (~46.7%), cellular processes (~35.1%), and single organism process (~34.3%) were high (Figure 4b).

The NK851 protein sequences were also compared with the evolutionary genealogy of genes through the *Non-supervised Orthologous Groups* (*eggNOG*) database by BLAST. *EggNOG* includes the biological orthologous gene clusters, and 4047 protein-coding genes were annotated. Among them, amino acid transport and metabolism genes (~9.9%), transcription genes (~8.2%), and general function prediction only genes (~10.1%) were relatively high (Figure 4c).

The *Kyoto Encyclopedia of Genes and Genomes* (*KEGG*) database was also applied for analyzing the genomic biological information related to biological pathways, diseases, drugs, and chemicals. A total of 1314 genes were compared (Figure 4d). In environmental information processing, the gene function with the largest proportion was the ABC transporters (~11.2%). In metabolism, biosynthesis of amino acid (~11.4%) and carbon metabolism functional genes (~10.5%) had high levels. In genetic information processing, ribosome-associated genes (~4.3%) accounted for the largest proportion.

### 2.4. Potassium-Solubilizing Mechanism of NK851

Acidic pH is supposed to be involved in potassium solubilization induced by potassium-solubilizing bacteria [23,24]. To investigate whether acidic pH contributed to the solubilization, the pH of the cultures was determined. With the increase in culturing time, the pH values of the NK851 cultures rapidly decreased from 7.0 to 4.8 in the first day and then was stable to 4 in the following days (Appendix A). Therefore, the condition of pH = 4 was used for potassium-release assays. Interestingly, similar to the condition of pH = 7, the condition of pH = 4 did not lead to remarkable increase in potassium release like in NK851 (Figure 5a). These results indicated that acidic pH was not involved in NK851-induced potassium release from potassium feldspar.

To further explore the mechanism of potassium release induced by NK851, the cultures of NK851 were used to prepare the culturing supernatant, the extracellular protein solution, and the protein-free supernatant for further potassium-release assays. Strikingly, after 3 days of treatment, the protein solution, similar to the initial culturing supernatants, led to drastic potassium release >100 mg/L. In contrast, the protein-free supernatant only induced slight potassium release up to 31 mg/L (Figure 5b). Therefore, the extracellular proteins secreted by NK851 played an important role in potassium release.

To further identify the potential extracellular proteins involved in potassium release, the proteins adsorbed by potassium feldspar after 12 h of co-incubation were separated by SDS-PAGE and identified by protein mass spectrometry. Two proteins were finally identified, i.e., the elongation factor TU and enolase. Alpha Fold2 prediction revealed that the elongation factor TU had both α-helixes and β-sheets in its three-dimensional structure (Figure 6a). Pymol prediction further showed that this protein exposed abundant side carboxyl groups of aspartic acid and glutamic acid (Figure 6b) and contained a carboxyl-group-rich cavity that may bind potassium feldspar via the hydrogen bonds between the carboxyl group and oxygen atoms on the surface of potassium feldspar (Figure 6c). Similarly, Alpha Fold2 and Pymol prediction of the enolase structure revealed abundant α-helixes (Figure 6d) together with exposure of abundant carboxyl groups and the potassium feldspar-binding cavity in this protein (Figure 6e,f). The formation of the strong hydrogen bonds between the two proteins and potassium feldspar is supposed to attenuate the interaction between the oxygen atoms and potassium, leading to release of potassium from potassium feldspar.

## 3. Discussion

In 1912, Passik first isolated a *Bacillus* strain from earthworm intestine, which could decompose aluminosilicate minerals such as feldspar and mica. In 1986, Avakyan established the taxonomic status of *Bacillus mucilaginosus* as a species through DNA-RNA homology analysis, 16S rRNA sequence analysis, and other ways [25], and this strain was discovered to have the ability to dissolve potassium. Nowadays, gene sequencing methods are diverse, and the cost is further reduced. It has been reported that many types of bacteria have the ability to solubilize potassium. Relatively mature potassium-solubilizing microorganisms have been put into production and application. In 2021, Raji et al. isolated saxicolous bacteria from saxicolous habitats and evaluated its promoting effect on tomato growth. In 2018, Pramanik isolated *Bacillus Pseudomycoides* O-5 from mica-waste-treated soil from northeast India, which increased the content of soluble potassium in soil and promoted the absorption of potassium by tea trees. In 2016, *Bacillus subtilis* ANctcri3 and *Bacillus megaterium* ANctcri7 were isolated from Kerala (India) to explore the possibility of using potassium bacteria as bio-fertilizers for tuber vegetables with high potassium demand [14,16,26]. Nowadays, researchers have begun to explore the mechanism of potassium-solubilizing bacteria. In this study, the highly efficient potassium-solubilizing bacterial strain, i.e., the *P. megaterium* strain NK851, was isolated from the *Astragalus sinicus* rhizosphere soil. After evaluating its potassium-solubilizing ability, the phylogenetic tree was constructed, and the whole genome was analyzed, which provided new genetic information of endospore-generating, potassium-solubilizing bacteria.

The potassium-solubilizing bacteria can dissolve the potassium-containing minerals and release potassium through a variety of mechanisms, such as acidolysis, indirect solubilization, and biofilm synthesis [27,28]. However, the potassium-solubilizing mechanisms remain to be further investigated. It is now agreed that this process is dominated by complex organic acids, but it is difficult to specify specific substances. In this study, we found that the main potassium-solubilizing mechanism of NK851 is not attributed to acidic pH. Furthermore, we identified two extracellular proteins binding potassium feldspar through mass spectrometry analysis and predicted the possible binding sites of potassium feldspar, which will provide new ideas for research on the potassium-solubilizing mechanism of potassium-solubilizing bacteria. This research supplies a theoretical basis for synthetic biological regulation and broadens the prospect of potassium-solubilizing bacteria as biological fertilizer. In the future researches, we will identify the genes encoding enolase and elongation factor TU and insert the genes into the expression vectors by using synthetic biology techniques such as de novo gene synthesis, seamless cloning, and large-fragment recombination. The obtained strains will be used to confirm the role of these two proteins in potassium solubilization.

Although people have been able to screen out high-efficiency potassium-solubilizing bacteria, these relatively mature bacterial fertilizers also have the disadvantages, e.g., weak stability in the soil environment, short shelf life, and long action period [24]. One critical reason contributing to the weak stability is that the potassium-solubilizing bacteria bind with difficulty and can be colonized at plant roots, and potassium-solubilizing bacteria have poor tolerance of the environment. Synthetic Microbial Communities (SynCom) is expected to provide solutions to this problem [29,30]. SynCom is composed of different microorganisms. Compared with a single strain, these microorganisms in a SynCom have a clearer division of labor and resource exchange, which are applied in many fields. Based on the bottom-up or top-down approaches of synthetic biology, researchers can improve the required biological production and the output of biological materials and realize multiple biological functions by artificially constructing microbial communities with defined behaviors [31,32,33]. In the future, based on the results of mass spectrometric analysis, we will use synthetic biology techniques for construction of potassium-solubilizing SynCom. For example, specific adhesion factors or artificial cells will be synthesized to enhance the adhesion of multiple potassium-solubilizing bacteria to plant roots, forming stable SynCom in the rhizosphere regions. This SynCom is expected to have enhanced tolerance to dry conditions, low temperatures, alkaline conditions, and other adversities so that it still has high potassium-solubilizing activity in the soil environment. Furthermore, pot experiments and field experiments will be conducted using the constructed potassium-solubilizing microbial system with cash crops as the material under the condition of using potassium feldspar as the only potassium source to determine soil fertility, rhizosphere microbial community structure, and aboveground physiological indicators (chlorophyll content, related enzyme metabolism level, etc.). Then, we will evaluate the remodeling of soil structures, and the improvement of the crop growth environment by the potassium-solubilizing microbial system may also be highlighted.

## 4. Materials and Methods

### 4.1. Isolation, Purification, and Culturing of Potassium-Solubilizing Bacteria

In order to screen and isolate potassium-solubilizing bacteria, the *Astragalus sinicus* rhizosphere soil was selected, which was sampled from a rice field in Hunan Province, China. First, 10 g of the soil sample was added into 90 mL of sterile water. The mixture was violently shaken to prepare well-dispersed soil suspension, which was further diluted with 10-fold gradients and spread on the potassium-solubilizing medium plates (sucrose 10 g, Na_2_HPO_4_ 0.4 g, (NH_4_)_2_SO_4_ 1 g, MgSO_4_ 0.1 g, FeCl_3_ 0.02 g, potassium feldspar powder 5 g, agar 20 g, and distilled water 1000 mL, pH = 7.2). The medium was cultured at 30 °C for 3~4 days. Single colonies were further purified on Luria–Bertani (LB) medium plates, obtaining the pure strains for further potassium-releasing assays.

### 4.2. Potassium-Releasing Assays

The purified strains were cultured in liquid LB medium overnight and harvested by centrifugation at 12,000 rpm for 2 min. The bacterial cells were then suspended in sterile water and added into 50 mL of liquid potassium-solubilizing medium to 10^7^ cells/mL. The mixtures were then cultured at 30 °C with shaking at 140 rpm for indicated days. At each day, 5 mL of the cultures were sampled and centrifuged at 12,000 rpm for 5 min. The potassium concentrations of the obtained supernatants were then determined by using an inductively coupled plasma (ICP) mass spectrometer (Perkin Elmer, Waltham, MA, USA) to evaluate the ability of the strains to release potassium from potassium feldspar.

### 4.3. Scanning Electron Microscopy (SEM)

To observe the cell morphology of NK851, the strain was cultured in the liquid LB medium at 30 °C with shaking for 1 day or in the liquid potassium-solubilizing medium for 3 days. The cells were harvested by centrifugation at 12,000 rpm for 1 min, fixed by 4% formaldehyde for 24 h, dehydrated by ethanol, and dried by a vacuum freeze drier (Scientz-10N, Ningbo Scientz Biotechnology, Ningbo, China). The samples were then observed by a scanning electron microscope (TESCAN MIRA LMS, Brno, Czech Republic).

### 4.4. Genomic DNA Extraction, 16S rDNA Sequencing, and Genome Sequencing of NK851

The freshly cultured *P. megaterium* NK851 cells were broken by ceramic and silica beads under vortexing treatment. The total genomic DNA was extracted by using a soil DNA extraction kit (Biomarker Technologies, Beijing, China), which led to high DNA yield and purity. The 16S rDNA of the strain was amplified from the total genomic DNA with the primers 27F (5′ AGAGTTTGATCCTGGCTCAG 3′) and 1492 R (5′ GGTTACCTTGTTACGACTT 3′). The purified polymerase chain reaction (PCR) product was further sequenced by Tsingke Biotechnology, Beijing, China. The 16S rDNA was further analyzed by the NCBI BLAST software (ElasticBLAST 1.0.0, https://blast.ncbi.nlm.nih.gov/doc/blast-news/2023-BLAST-News.html, accessed on 1 March 2023) and the phylogenetic tree of the strain was constructed by the MEGA11.0 software with the neighbor-joining method [34].

For genome sequencing, the high-quality genome DNA of NK851 was purified by BluePippin automatic DNA recycling system. The DNA library was further constructed by using the SQK-LSK109 ligation kit for genomic sequencing (Biomarker Technologies, Beijing, China). The obtained sequences were filtered to assemble the genome by using Canu v1.5 software, and the assembled genome was cyclized by the circulator v1.5.5 software. The genome component prediction was performed by Prodigal v2.6.3, GeneWise v2.2.0, PhiSpy v2.3, antiSMASH v5.0.0, and PromPredict v1. Functional annotation was performed by using the NR, Swiss-Prot, TrEMBL, KEGG, and eggNOG, and Blast2go tools (http://www.biocloud.net, assessed on 1 March 2023).

### 4.5. Exploration of Potassium-Solubilizing Mechanism in NK851

To investigate whether acidic pH contributes to potassium solubilization, three groups were set, i.e., the pH = 7 group in which neither the strain nor HCl was added into the potassium-solubilizing medium, the pH = 4 group in which HCl was added to pH = 4, and the NK851 group in which only the *P. megaterium* NK851 cells was added to 1 × 10^7^ cells/mL. After incubation at 30 °C with shaking for indicated time, the cultures were sampled and centrifuged at 120,000 rpm for 5 min. The potassium contents in the supernatants were measured by the ICP mass spectrometer.

To evaluate the contribution of the NK851 extracellular proteins and metabolic small molecules to potassium solubilization, the NK851 cells were cultured in liquid potassium-solubilizing medium for 3 days, followed by centrifugation at 12,000 rpm for 2 min, thus obtaining the supernatant. The total proteins in the supernatant were isolated by using ultrafiltration tubes (cut-off weight = 3000). The proteins were then dissolved in fresh liquid potassium-solubilizing medium, thus obtaining the protein solution. The ultrafiltration liquid was also collected as the protein-free supernatant. Five groups were set, i.e., the control group with only the potassium-solubilizing medium, the NK851 group with NK851 cells added into the medium to 1 × 10^7^ cells/mL, the supernatant group in which potassium feldspar was added into the supernatant to 5 g/L, the protein solution group in which potassium feldspar was added into the protein solution to 5 g/L, and the protein-free supernatant in which potassium feldspar was added into the protein-free supernatant to 5 g/L. The mixtures were incubated at 30 °C with shaking for 3 days, followed by centrifugation and measurement of potassium contents in the supernatants.

### 4.6. Protein Mass Spectrometry and Structure Prediction

To identify potential proteins of NK851 involved in potassium solubilization, 0.5 g of potassium feldspar was added into 100 mL of the protein solution obtained in the Section 4.5. The mixture was gently shaken at 60 rpm for 12 h and then centrifuged at 12,000 rpm for 10 min. The proteins adsorbed by the pellets were detached by using 5 × SDS PAGE loading buffer at 95 °C for 5 min and separated by SDS-PAGE. The protein bands were cut for mass spectrometry (Beijing Protein Innovation, Beijing, China). Alpha Fold2 [35] and Pymol (The PyMOL Molecular Graphics System, Version 2.0 Schrödinger, LLC.) were used for structure prediction of the identified proteins.

### 4.7. Statistical Analysis

Each experiment was performed with three replicates (i.e., *n* = 3), and the values are shown with the means ± the standard errors. Differences between the groups were compared by Student’s *t*-tests (*p* < 0.05). All statistical tests were performed using the SPSS software package (Version 20, IBM, Armonk, NY, USA).

## 5. Conclusions

In conclusion, this study screened the potassium-solubilizing bacterium NK851 from rhizosphere soil, which efficiently released potassium from potassium feldspar. Further characterization revealed that it belonged to the *P. megaterium* species, strongly binding potassium feldspar and releasing potassium in a time-dependent manner. Genome analysis showed that NK851 has abundant genes involved in potassium decomposition and utilization, e.g., adsorption to mineral potassium, potassium release, and intracellular trafficking. Moreover, the strong potassium-releasing capacity of NK851 is not attributed to the acidic pH but is attributed to the extracellular proteins, such as the elongation factor TU and enolase that contain potassium feldspar-binding cavities. This study provides new information for exploration of the bacterium-mediated potassium solubilization mechanisms and for construction of synthetic organisms as high-efficient and stable fertilizers in agricultural application.

## Figures and Tables

**Figure 1 ijms-24-14226-f001:**
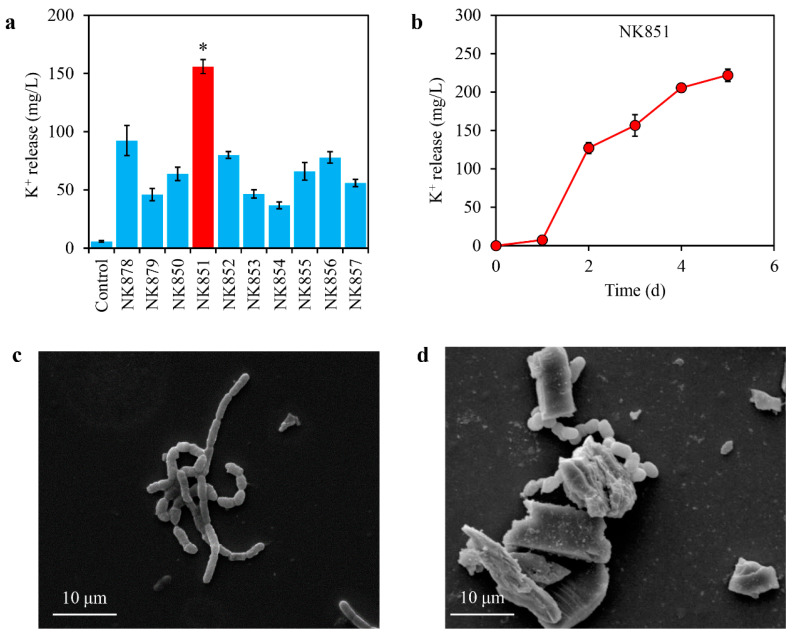
Potassium-solubilizing capacity of the isolated strains and morphology of the strain NK851. (**a**) Potassium release induced by the isolated strains. (The red column represents the strain NK851 with the highest potassium-solubilizing capacity) (**b**) Time-dependent potassium release induced by NK851. (**c**) The SEM image of NK851 in LB liquid medium. (**d**) The SEM image of NK851 cultured in the potassium-solubilizing medium. The asterisk (*) indicates significant difference between NK851 and the other strains (*p* < 0.05).

**Figure 2 ijms-24-14226-f002:**
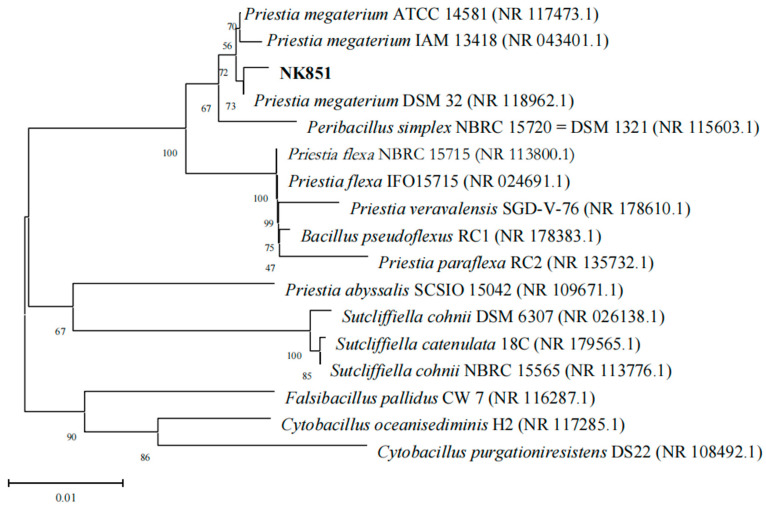
The phylogenetic tree of NK851 constructed by the MEGA11.0 software with the neighbor-joining method.

**Figure 3 ijms-24-14226-f003:**
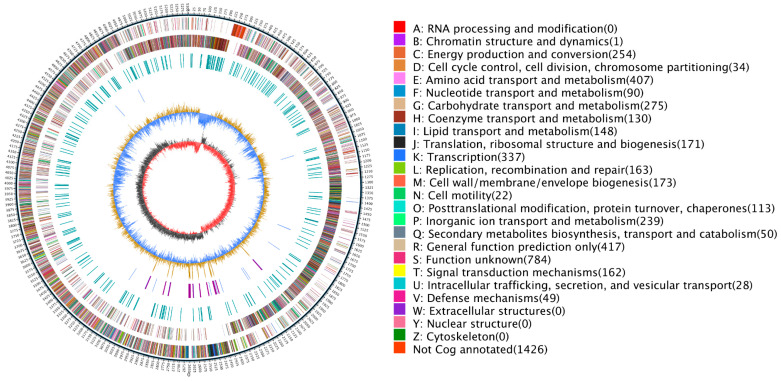
The genome-wide circle map of the *P. megaterium* strain NK851. There are six circle areas in the chart, which represent the genome size (each scale is 5 kb), genome positive-strand genes, genome negative-strand genes, tRNA (blue) and rRNA (purple), GC content, and GC-skew from outside to inside. Among the positive and negative genes in the genome, different colors represent different COG functional classifications, and the specific classification is shown on the right side of the figure. In the GC content, light yellow (blue) indicates that the GC content in this region is higher (lower) than the average GC content of the genome, and the higher the peak value, the greater the difference from the average GC content.

**Figure 4 ijms-24-14226-f004:**
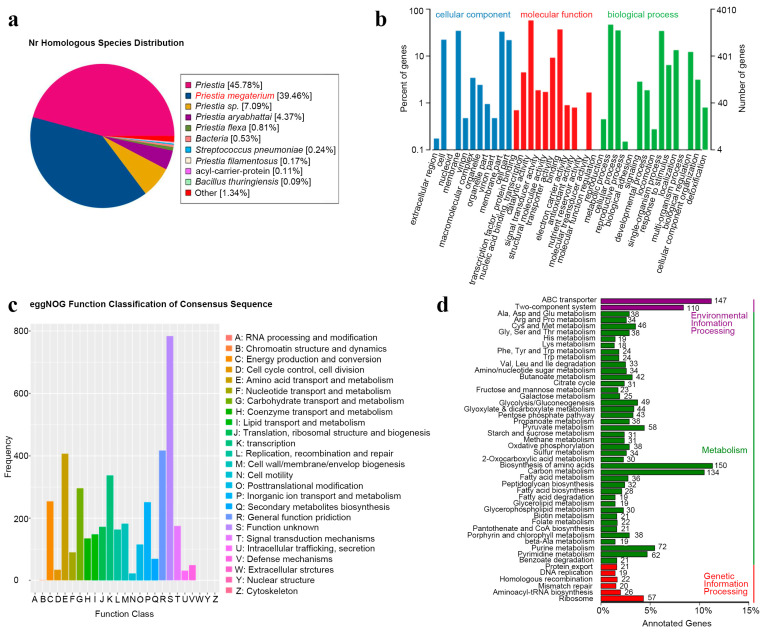
Genome analysis of the *P. megaterium* strain NK851. (**a**) Species distribution map of the genome compared to the *NR* database. (**b**) Statistical chart of the GO function annotation classes. The abscissa is the classification of different functional genes in the GO database. The left side of the ordinate is the percentage of the number of different functional genes, and the right side is the number of genes. (**c**) Statistical chart of the functional class of eggNOG genes. The abscissa is the classification of different functional genes in the eggNOG database. (**d**) Statistical chart of the KEGG annotation classes. The ordinate is the classification of different functional genes in the KEGG database.

**Figure 5 ijms-24-14226-f005:**
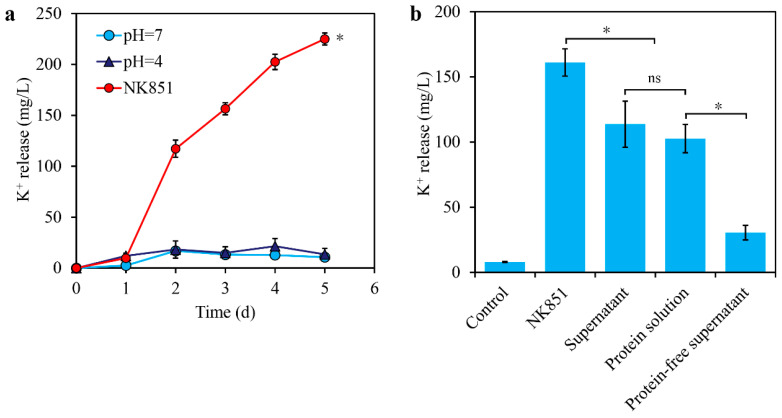
Effect of pH (**a**), NK851, and the metabolites of NK851 on potassium release from potassium feldspar. (**a**) Potassium release from potassium feldspar in the medium with different pH values or in the NK851 cultures. (**b**) Potassium release from potassium feldspar induced by NK851, NK851 culturing supernatant, extracellular protein solution, and protein-free supernatant. The asterisk (*) indicates significant difference between the groups, while “ns” indicates no significant difference (*p* < 0.05).

**Figure 6 ijms-24-14226-f006:**
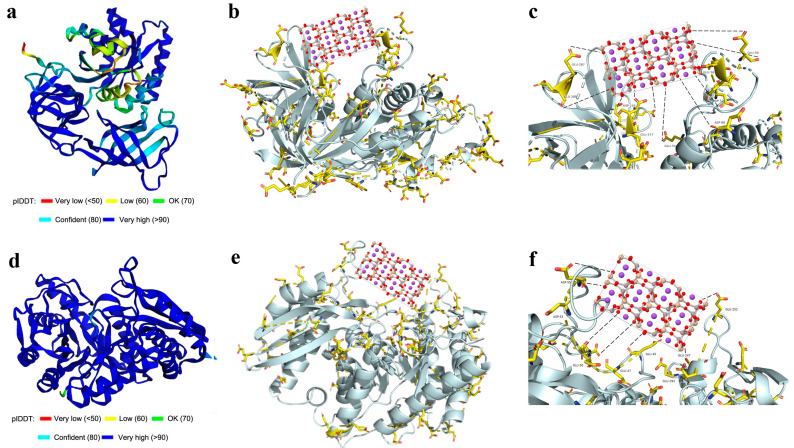
Structure prediction of the identified potassium feldspar-binding proteins elongation factor TU (**a**–**c**) and enolase (**d**–**f**). (**a**) Structure of the identified elongation factor TU predicted by Alpha Fold2. The different colors represent different pLDDT; (**b**,**c**) the structure of TU predicted by Pymol. The images indicate the interaction between the acidic amino acid groups and potassium feldspar. The purple and red balls represent potassium atoms and oxygen atoms, respectively. The dotted lines indicate the interaction between the acidic groups exposed on the protein surface and the oxygen atoms that may lead to potassium atoms. (**d**) Structure of the identified enolase predicted by Alpha Fold2. (**e**,**f**) The structure of TU predicted by Pymol.

## Data Availability

The data that support the findings of this study are not openly available due to reasons of sensitivity and are available from the corresponding author upon reasonable request. Data are located in controlled access data storage at Nankai University.

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
