# Peer review of "Analysis of the Potassium-Solubilizing Priestia megaterium Strain NK851 and Its Potassium Feldspar-Binding Proteins"

_ijms, 2023, doi:10.3390/ijms241814226_

Round 1

Reviewer 1 Report

This article titled “Analysis of the potassium-solubilizing Priestia megaterium strain NK851 and its potassium feldspar-binding proteins” mainly reported the new bacteria  exploration of potassium-solubilizing bacteria as microbial fertilizers. During ongoing efforts to search for new bioactive bacteria a high-efficiency potassium-solubilizing bacterium, named NK851, were isolated from the Astragalus sinicus rhizosphere soil. 

The presented manuscript will certainly be of interest to readers and researchers working in the fields of microbiology, biochemistry  and biotechnology.

Minor comments:

It would be interesting if a possible biosynthesis of the extracellular proteins involved in potassium release of Priestia megaterium  strain NK851 were discussed in some detail in discussion part. 

Author Response

Response to Reviewer 1

Comments and Suggestions for Authors

This article titled “Analysis of the potassium-solubilizing Priestia megaterium strain NK851 and its potassium feldspar-binding proteins” mainly reported the new bacteria exploration of potassium-solubilizing bacteria as microbial fertilizers. During ongoing efforts to search for new bioactive bacteria a high-efficiency potassium-solubilizing bacterium, named NK851, were isolated from the Astragalus sinicus rhizosphere soil. The presented manuscript will certainly be of interest to readers and researchers working in the fields of microbiology, biochemistry and biotechnology.

Response:

Thank you very much for your positive comments and important suggestions to our manuscript. We have carefully revised the paper based on your kind suggestions.

Minor comments:

It would be interesting if a possible biosynthesis of the extracellular proteins involved in potassium release of Priestia megaterium strain NK851 were discussed in some detail in discussion part. 

Response:

According to your kindly suggestion, the biosynthesis of the extracellular proteins involved in potassium release was discussed (Page 9, Lines 260~264):

“In the future research, we will identify the genes encoding enolase and elongation factor TU, and insert the genes into the expression vectors by using synthetic biology techninques, such as de novo gene synthesis, seamless cloning and large fragment recombination. The obtained strains will be used to confirm the role of these two proteins in potassium solubilization.”

Reviewer 2 Report

In this study a potassium-solubilizing bacterium belonging to the species Priestia megaterium was isolated from the Astragalus sinicus rhizosphere soil. This bacterium was able to grow in the medium with potassium feldspar as the sole potassium source. Taxonomic affiliation was determined by 16S rDNA sequencing. Sequencing of the whole genome was carried out to identify genes involved in adherence to mineral potassium, potassium release, and intracellular trafficking. Results revealed that the potassium-releasing capacity of NK851 can be attributed to the extracellular potassium feldspar-binding proteins, such as the elongation factor TU and the enolase that contain potassium feldspar-binding cavities.

General comment

The research on phosphate solubilizing bacteria is interesting because it can contribute to the development bio-based products for a more sustainable agriculture. The introduction, the description of the methods and the results are enough clear while the discussion section should be rewritten because is less focused on the interpretation of the results obtained in this study. There is some confusion on the use of synthetic biology to set up microbial communities with improved plant growth promoting activity. The review recently published by Shayanthan and colleagues (doi: 10.3389/fagro.2022.896307), could help to clarify this point.

Specific comments

Line 77: Please change “After co-incubation of” to “After inoculation in”.

Lines 81-82. Please eliminate “and therefore was determined as the high-efficiency potassium-solubilizing strain in this study.”

Lines 82-83. Please check the grammar of “With the increase in the culturing time of NK851, the released potassium levels slightly increased to 7 mg/L after 1 day, and then remarkably increased to 127 mg/L after 2 days, and reached 222 mg/L after 5 days”.

Line 90. Please eliminate “(a, b)”.

Was the sequence of 16S rDNA gene of the strain NK851 deposited in public database?

Line 114. Change “repeat” to “repeated”.

Line 148. Which strain of P. megaterium? Was the type strain?

Figure 4 is less readable.

Lines 162-165. Could you indicate some value in percentage?

Lines 176-178. Could you indicate some value in percentage?

Line 185: Please change “trended” to “was stable to”

Lines 185-198. The explanation provided is non convincing because at acidic pH usually the bacteria stop to grow. Could you please explain better your idea?

Lines 230-245. The hystory of bacteria belonging to this species of solubilizing bacteria is too long for the discussion section.

Line 243. Kerala is a state of India that is a nation.

Line 243-244. Please clarify the sentence “and determined it as an early identification of the strain with the ability to solubilize potassium”.

Lines 247-251. This part concerns the metodology carried out in this study. You must comment the findings. I suggest eliminating this part.

Lines 258-261. You have to comment these findings also in relation if possible, to previous studies.

Lines 268-280. This part is not clear, indeed, the approach described is not the definition of synthetic biology to set up microbial communities with improved plant growth promoting activity. See the review by Shayanthan et al. (2022) The Role of Synthetic Microbial Communities (SynCom) in Sustainable Agriculture. Front. Agron. 4:896307. doi: 10.3389/fagro.2022.896307

Line 300. What do you mean for "activated"?

Line 318. The kit is usually used to extract DNA from soil samples not from pure cultures. Please explain why this kit was used.

Line 334. Please add “P. megaterium” before “NK851”.

Line 373. Add “species” after “megaterium”.

Lines 373-374. Please check the grammar.

Grammar of the manuscript shoul be checked to make it more clear to the reader.

Author Response

Response to Reviewer 2

Comments and Suggestions for Authors

In this study a potassium-solubilizing bacterium belonging to the species Priestia megaterium was isolated from the Astragalus sinicus rhizosphere soil. This bacterium was able to grow in the medium with potassium feldspar as the sole potassium source. Taxonomic affiliation was determined by 16S rDNA sequencing. Sequencing of the whole genome was carried out to identify genes involved in adherence to mineral potassium, potassium release, and intracellular trafficking. Results revealed that the potassium-releasing capacity of NK851 can be attributed to the extracellular potassium feldspar-binding proteins, such as the elongation factor TU and the enolase that contain potassium feldspar-binding cavities.

General comment

The research on phosphate solubilizing bacteria is interesting because it can contribute to the development bio-based products for a more sustainable agriculture. The introduction, the description of the methods and the results are enough clear while the discussion section should be rewritten because is less focused on the interpretation of the results obtained in this study. There is some confusion on the use of synthetic biology to set up microbial communities with improved plant growth promoting activity. The review recently published by Shayanthan and colleagues (doi: 10.3389/fagro.2022.896307), could help to clarify this point.

Response:

Thank you very much for your positive comments and important suggestions to our manuscript. We have carefully revised the paper based on your kind suggestions. Especially, we rewrote the discussion part to more focus on the results, and cited the paper you kindly recommended (Page 10, Lines 293~306):

“Synthetic Microbial Communities (SynCom) is expected to provide solutions to this problem [29, 30]. SynCom is composed of different microorganisms. Compared with a single strain, these microorganisms in a SynCom have a clearer division of labor and resource exchange, which are applied in many fields. Based on the bottom-up or top-down approaches of synthetic biology, researchers can improve the required biological production and the output of biological materials and realize multiple biological functions by artificially constructing microbial communities with defined behaviors [31-33]. In the future, based on the results of mass spectrometric analysis, we will use synthetic biology techniques for construction of potassium-solubilizing SynCom. For example, specific adhesion factors or artificial cells will be synthesized to enhance the adhesion of multiple potassium-solubilizing bacteria to plant roots, forming stable SynCom in the rhizosphere regions. This SynCom is expected to have enhanced tolerance to dry, low temperature, alkaline and other adversities, so that it still has high potassium-solubilizing activity in the soil environment.”

Specific comments

Line 77: Please change “After co-incubation of” to “After inoculation in”.

Response:

The sentence has been revised (Page 2, Line 76):

“After incubation in the strains and the liquid potassium-solubilizing medium for 3 days, …”

Lines 81-82. Please eliminate “and therefore was determined as the high-efficiency potassium-solubilizing strain in this study.

Response:

The content “and therefore was determined as the high-efficiency potassium-solubilizing strain in this study.” has been eliminated (Page 2, Lines 78~79):

“Especially, the strain NK851 induced the highest levels of potassium release (156 mg/L, Fig. 1a).”

Lines 82-83. Please check the grammar of “With the increase in the culturing time of NK851, the released potassium levels slightly increased to 7 mg/L after 1 day, and then remarkably increased to 127 mg/L after 2 days, and reached 222 mg/L after 5 days”.

Response:

We have changed “With the increase in the culturing time of NK851” to “As the culture time of NK851 increased” (Page 2, Lines 79~81):

“As the culture time of NK851 increased, the released potassium levels slightly increased to 7 mg/L after 1 day, and then remarkably increased to 127 mg/L after 2 days, and reached 222 mg/L after 5 days (Fig. 1b).”

Line 90. Please eliminate “(a, b)”.

Response:

The “(a, b)” has been eliminated from the sentence (Page 3, Line 88):

Figure 1. Potassium-solubilizing capacity of the isolated strains and morphology of the strain NK851.”

Was the sequence of 16S rDNA gene of the strain NK851 deposited in public database?

Response:

The sequence of 16S rDNA gene is submitting to the public database.

Line 114. Change “repeat” to “repeated”.

Response:

The word “repeat” has been changed to “repeated” (Page 4, Line 114):

“There are 11 Clustered Regularly Interspaced Short Palindromic Repeated (CRISPR) sequences and 4 gene islands in the genome.”

Line 148. Which strain of P. megaterium? Was the type strain?

Response:

The information of the standard strain has been supplied (Page 5, Lines 146~147):

“A total of 5,314 protein-encoding genes were annotated. Combined with the comparison results of phylogenetic trees, it was finally concluded that NK851 has the highest similarity with the P. megaterium standard strain 22-2 (Reference Genome Number ASM993541v1, 39.46%, Fig. 4a), and it was determined as P. megaterium.”

Figure 4 is less readable.

Response:

The Figure 4 has been redrawn for clearer reading (Page 6, Line 148).

Lines 162-165. Could you indicate some value in percentage?

Response:

We already added some values in the sentence (Page 6, Line 159~163):

“In the cell components, the membrane-associated protein-coding genes were the most abundant (~34.3%). In the molecular function, the content of protein-coding genes related to catalysis (~57.5%) and binding (~36.9%) is high. In biological processes, the protein-coding genes related to metabolism (~46.7%), cellular processes (~35.1%), and single organism process (~34.3%) are high (Fig. 4b).”

Lines 176-178. Could you indicate some value in percentage?

Response:

We already added some values in the sentence (Page 7, Lines 172~176):

“In the environmental information processing, the gene function with the largest proportion was the ABC transporters (~11.2%). In metabolism, biosynthesis of amino acid (~11.4%), carbon metabolism functional genes (~10.5%) had a high level. In genetic information processing, ribosome-associated genes (~4.3%) accounted for the largest proportion.”

Line 185: Please change “trended” to “was stable to”

Response:

The word “trended” has been changed to “was stable to” (Page 7, Line 183):

“With the increase in culturing time, the pH values of the NK851 cultures rapidly de-creased from 7.0 to 4.8 in the first day, and then was stable to 4 in the following days (Fig. S1).”

Lines 185-198. The explanation provided is non convincing because at acidic pH usually the bacteria stop to grow. Could you please explain better your idea?

Response:

We are sorry that the experimental procedures were not introduced in accuracy in the previous legend of Figure 5. The experiment in Figure 5a is to investigate the effect of pH on potassium release. In fact, both the pH=7 condition and the pH=4 condition in Figure 5a did not include culturing of NK851. We found that, similar to the condition of pH = 7, the condition of pH = 4 did not led to remarkable increase in potassium release like NK851 (Fig. 5a). These results indicated that acidic pH was not involved in NK851-induced potassium release from potassium feldspar. We revised the figure legend of Figure 5a for more accuracy (Page 7, Lines 197~199):

Figure 5. Effect of pH (a), NK851, and the metabolites of NK851 on potassium release from potassium feldspar. (a) Potassium release from potassium feldspar in the medium with different pH values or in the NK851 cultures.”

Lines 230-245. The history of bacteria belonging to this species of solubilizing bacteria is too long for the discussion section.

Response:

The introduction of the history of bacteria has been simplified (Page xx, Line xx):

“In 1912, Passik first isolated a Bacillus strain from the earthworm intestine, which could decompose aluminosilicate minerals such as feldspar and mica. In 1986, Avaky-an established the taxonomic status of Bacillus mucilaginosus as a species through DNA-RNA homology analysis, 16s rRNA sequence analysis and other ways [25], and determined it as an early identification of the strain with the ability to solubilize potassium.”

Line 243. Kerala is a state of India that is a nation.

Response:

We have corrected the sentence (Page 9, Line 241):

“In 2016, Bacillus subtilis ANctcri3 and Bacillus megaterium ANctcri7 were isolated from Kerala of India to explore the possibility of using potassium bacteria as bio-fertilizers for tuber vegetables with high potassium demand [14, 26, 16].”

Line 243-244. Please clarify the sentence “and determined it as an early identification of the strain with the ability to solubilize potassium”.

Response:

We have revised the sentence to make it more readable (Page 8, Lines 232~233):

“In 1986, Avakyan established the taxonomic status of Bacillus mucilaginosus as a species through DNA-RNA homology analysis, 16S rRNA sequence analysis and other ways [25], and this strain is an early discovered strain with the ability to dissolve potassium.”

Lines 247-251. This part concerns the metodology carried out in this study. You must comment the findings. I suggest eliminating this part.

Response:

We have eliminated this part (Page 9, Line 244).

Lines 258-261. You have to comment these findings also in relation if possible, to previous studies.

Response:

We have commented this study in relation to previous studies (Page 9, Lines 240~248):

“In 2016, Bacillus subtilis ANctcri3 and Bacillus megaterium ANctcri7 were isolated from Kerala of India to explore the possibility of using potassium bacteria as bio-fertilizers for tuber vegetables with high potassium demand [14, 26, 16]. Nowadays, researchers begin to explore the mechanism of potassium-solubilizing bacteria. In this study, the highly efficient potassium-solubilizing bacterial strain, i.e., the P. megaterium strain NK851, was isolated from the Astragalus sinicus rhizosphere soil. After evaluating its potassium-solubilizing ability, the phylogenetic tree was constructed and the whole genome was analyzed, which provided new genetic information of endospore-generating potassium-solubilizing bacteria.”

Lines 268-280. This part is not clear, indeed, the approach described is not the definition of synthetic biology to set up microbial communities with improved plant growth promoting activity. See the review by Shayanthan et al. (2022) The Role of Synthetic Microbial Communities (SynCom) in Sustainable Agriculture. Front. Agron. 4:896307. doi: 10.3389/fagro.2022.896307

Response:

We have figured out the related concepts with this article, and cited the reference you kindly suggested (Page 10, Lines 293~306):

“Synthetic Microbial Communities (SynCom) is expected to provide solutions to this problem [29, 30]. SynCom is composed of different microorganisms. Compared with a single strain, these microorganisms in a SynCom have a clearer division of labor and resource exchange, which are applied in many fields. Based on the bottom-up or top-down approaches of synthetic biology, researchers can improve the required biological production and the output of biological materials and realize multiple biological functions by artificially constructing microbial communities with defined behaviors [31-33]. In the future, based on the results of mass spectrometric analysis, we will use synthetic biology techniques for construction of potassium-solubilizing SynCom. For example, specific adhesion factors or artificial cells will be synthesized to enhance the adhesion of multiple potassium-solubilizing bacteria to plant roots, forming stable SynCom in the rhizosphere regions. This SynCom is expected to have enhanced tolerance to dry, low temperature, alkaline and other adversities, so that it still has high potassium-solubilizing activity in the soil environment.”

Line 300. What do you mean for "activated"?

Response:

The “activated” means “cultured”. The word has been replaced (Page 10, Line 328):

“The purified strains were cultured in liquid LB medium overnight and harvested by centrifugation at 12,000 rpm for 2 min.”

Line 318. The kit is usually used to extract DNA from soil samples not from pure cultures. Please explain why this kit was used.

Response:

The kit was used since it . The reason has been introduced in this part (Page 11, Lines 346~347):

“The total genomic DNA was extracted by using a soil DNA extraction kit (Biomarker Technologies, Beijing, China), which led to high DNA yield and high DNA purity.”

Line 334. Please add “P. megaterium” before “NK851”.

Response:

We have added “P. megaterium” before “NK851” (Page 11, Line 366):

  “…, and the NK851 group in which only the P. megaterium NK851 cells was added to 1×107 cells/mL.”

Line 373. Add “species” after “megaterium”.

Response

We have added “species” after “megaterium” (Page 12, Line 401):

“Further characterization revealed that it belonged to the P. megaterium species, strongly binding potassium feldspar and release potassium in a time-dependent manner.”

Lines 373-374. Please check the grammar.

Response:

We have revised this sentence (Page 12, Lines 402~405):

“Genome analysis showed that NK851 possesses abundant genes involved in potassium decomposition and utilization, e.g., adsorption to mineral potassium, potassium release, and intracellular trafficking.”

Round 2

Reviewer 2 Report

I appreciate all the efforts of the authors to fulfill the requirements of the reviewer toimprove the manuscript. After carefully reading the revised manuscript, I found it was improved, however, there are important issues that still need to be clarified. See specific comments below.

Lines 97-98. P. megaterium ATCC 14581 and P. megaterium NBRC 15308 are the same strain. Indeed, the type strain P. megaterium ATCC 14581 has different strain designation. Please correct also the phylogenetic tree (Figure 2).

Line 147.What do you mean for “standard strain? The type strain of P. megaterium is P. megaterium ATCC 14581.

Please provide the accession number of the deposited sequence of P. megaterium NK851.

Line 234. Please change “and this strain is an early discovered strain with the ability to dissolve potassium. “ to “and this strain was discovered to have the ability to dissolve potassium.” Do you mean that this strain was the first described strain able to dissolve potassium? Which was the strain name?

Line 243. Please change “were isolated from Kerala of India” to “were isolated from Kerala (India)”.

Lines 348-349. Please change “which led to high DNA yield and high DNA purity.” to which led to high DNA yield and purity”.

Line 405. Please “possesses” to “has”.

Moderate changes of the english language are required. 

Author Response

Response to Reviewers

I appreciate all the efforts of the authors to fulfill the requirements of the reviewer to improve the manuscript. After carefully reading the revised manuscript, I found it was improved, however, there are important issues that still need to be clarified. See specific comments below.

Response:

Thank you very much again for your kind suggestions. We re-revised the manuscript based on your comments, and responded to your comments point by point.

Lines 97-98. P. megaterium ATCC 14581 and P. megaterium NBRC 15308 are the same strain. Indeed, the type strain P. megaterium ATCC 14581 has different strain designation. Please correct also the phylogenetic tree (Figure 2).

Response:

The strain number of ATCC 14581 has been consistent, and the Figure 2 is revised. Please see the Manuscript (Page 3, Lines 96~97):

“The results indicated that its 16S rDNA sequence has the highest similarity with those of the P. megaterium ATCC 14581 (98.83%) and the P. megaterium IAM 13418 (98.64%).”

Figure 2. The phylogenetic tree of NK851 constructed by the MEGA11.0 software with the Neighbor-Joining method.

Line 147.What do you mean for “standard strain? The type strain of P. megaterium is P. megaterium ATCC 14581.

Response:

The reference genome is the P. megaterium strain 22-2 (Reference Genome Number ASM993541v1). Therefore, the word “standard” has been replaced by “reference” (Page 5, Line 146):

“Combined with the comparison results of phylogenetic trees, it was finally concluded that NK851 has the highest similarity with the P. megaterium reference strain 22-2 (Reference Genome Number ASM993541v1, 39.46%, Fig. 4a), …”

Please provide the accession number of the deposited sequence of P. megaterium NK851.

Response:

The accession number of the deposited sequence of the strain has been provided in the Manuscript (Page 3, Line 95):

“In order to assure the taxonomic position of NK851, BLAST analysis was performed based on the sequencing results of NK851 16S rRNA (Accession number PRJNA1008360).”

Line 234. Please change “and this strain is an early discovered strain with the ability to dissolve potassium. “ to “and this strain was discovered to have the ability to dissolve potassium.” Do you mean that this strain was the first described strain able to dissolve potassium? Which was the strain name?

Response:

The sentence has been revised. In the initial paper, the strain had no specific name. The revised sentence in the Manuscript (Page 8, Lines 232~233):

“In 1986, Avakyan established the taxonomic status of Bacillus mucilaginosus as a species through DNA-RNA homology analysis, 16S rRNA sequence analysis and other ways [25], and this strain was discovered to have the ability to dissolve potassium.”

Line 243. Please change “were isolated from Kerala of India” to “were isolated from Kerala (India)”.

Response:

The sentence has been revised (Page 9, Line 241):

“In 2016, Bacillus subtilis ANctcri3 and Bacillus megaterium ANctcri7 were isolated from Kerala (India) to explore the possibility of using potassium bacteria as bio-fertilizers for tuber vegetables with high potassium demand [14, 26, 16].”

Lines 348-349. Please change “which led to high DNA yield and high DNA purity.” to “which led to high DNA yield and purity”.

Response:

The sentence has been revised (Page 11, Lines 346~347):

“The total genomic DNA was extracted by using a soil DNA extraction kit (Biomarker Technologies, Beijing, China), which led to high DNA yield and purity.”

Line 405. Please “possesses” to “has”.

Response:

The word has been replaced to “has” (Page 12, Line 403):

  “Genome analysis showed that NK851 has abundant genes involved in potassium decomposition and utilization, e.g., adsorption to mineral potassium, potassium release, and intracellular trafficking.”